# Tropical Red Fruit Blend Foam Mat Drying: Effect of Combination of Additives and Drying Temperatures

**DOI:** 10.3390/foods12132508

**Published:** 2023-06-28

**Authors:** Yaroslávia Ferreira Paiva, Rossana Maria Feitosa de Figueirêdo, Alexandre José de Melo Queiroz, Lumara Tatiely Santos Amadeu, Carolaine Gomes dos Reis, Francislaine Suelia dos Santos, Antônio Gilson Barbosa de Lima, Wilton Pereira da Silva, Josivanda Palmeira Gomes, Daniela Dantas de Farias Leite, Thalis Leandro Bezerra de Lima

**Affiliations:** 1Science and Technology Center, Federal University of Campina Grande, Campina Grande 58429-900, Brazil; antonio.gilson@ufcg.edu.br; 2Department of Agricultural Engineering, Federal University of Campina Grande, Campina Grande 58429-900, Brazil; rossanamff@gmail.com (R.M.F.d.F.); alexandrejmq@gmail.com (A.J.d.M.Q.); lumaratatielyea@gmail.com (L.T.S.A.); carolainetecalimentos@gmail.com (C.G.d.R.); francislainesuelis@gmail.com (F.S.d.S.); wiltonps@uol.com.br (W.P.d.S.); josivanda@gmail.com (J.P.G.); danieladantasfl@gmail.com (D.D.d.F.L.); tthalisma@gmail.com (T.L.B.d.L.)

**Keywords:** *Malpighia emarginata*, *Psidium guajava*, *Eugenia uniflora*, gum Arabic, gelatin, guar gum

## Abstract

Foam mat drying is a widely used technique for liquid products because it has a number of advantages; however, for an efficient process, the choice of additives and temperatures is extremely important. The objective of this study was to evaluate the effect of additives and drying temperatures on the powders obtained from the blend of tropical red fruits, such as acerola, guava, and pitanga. The foam formulations were prepared by mixing the pulps of the three fruits in equal proportions (1:1:1), all added with 6% albumin and 1% stabilizing agent: E1, gum Arabic; E2, guar gum; E3, gelatin. The combinations were subjected to beating, and subsequently, they were dried in an oven with forced air circulation at four temperatures (50 to 80 °C), with a mat thickness of 0.5 cm. The obtained powders showed low levels of water and water activity and high levels of bioactive compounds, colors with a predominance of yellow, intermediate cohesiveness, poor fluidity, and solubility above 50%. The best temperature for obtaining the powders was 60 °C. The formulation that produced the best results for the production of the tropical red fruit blend powder was the combination of albumin and gelatin.

## 1. Introduction

Acerola is an excellent source of vitamin C and bioactive compounds, such as phenolic compounds and anthocyanins, which have antioxidant properties [1]. Pitanga also has excellent nutritional value, with high concentrations of provitamin A, vitamin C, calcium, phosphorus, and phenolic compounds [2]. Guava has excellent acceptance for consumption in its most diverse forms, a fundamental requirement for inclusion in the daily diet of consumers, in addition to presenting minerals, carotenoids, ascorbic acid, and polyphenols, and being an important source of lycopene [3,4,5].

The food market undertakes a constant search for innovations aiming to attract an increasing number of buyers who may be interested in products with excellent nutritional characteristics, without giving up on flavor and practicality in the preparation. The association of fruits in the form of a blend provides the creation of a food that can overcome the nutritional characteristics of existing products, in addition to originating unique flavors, aromas, and colors.

The high water content of fruits and their derivatives, however, with their characteristic propensity to rapid deterioration, requires the adoption of treatments to provide an increase in shelf-life, with drying being one of the most used. Spray drying, convective drying, lyophilization, foam mat drying, and microwave drying are the most used methods to produce powdered products [6], which in addition to solving the problem of the shelf-life, also circumvent difficulties related to logistics costs by causing reductions in mass and volume and enhancing the concentration of active ingredients.

Powders produced by foam mat drying can be easily reconstituted and produced at a lower cost compared to other techniques. In addition, they are quite effective in retaining their nutritional characteristics [7,8]. However, for liquid foods to become foams, additives are needed that help this process, and they can be divided into foaming and stabilizing agents [9]. Consequently, for efficient drying, the choice of additives and temperatures is extremely important, directly affecting the quality of the powders.

Proteins are considered good foaming agents, and albumin has been widely used due to its gelling and foaming properties. However, foams are unstable by nature and there is a release of energy during relaxation, most often requiring the addition of a stabilizing agent, which provides stability under the heat and support of its structure, at least initially, since any collapse in the structure can result in inefficient drying [9].

Since it is an extremely important parameter, several studies have been carried out evaluating the behavior of different stabilizers in different foods.

Countless products have been obtained by foam mat drying using albumin, gum Arabic, guar gum, and gelatin as adjuvants, but there are no studies with the blend of acerola, guava, and pitanga. Thus, this study was carried out with the objective of evaluating the effect of the combination of albumin and gum Arabic additives, albumin and guar gum, albumin and gelatin, and the temperatures of 50, 60, 70, and 80 °C, in the foam mat drying of the blend of acerola, guava, and pitanga, determining the physicochemical parameters, bioactive compounds, physical properties, and colorimetry.

## 2. Materials and Methods

### 2.1. Raw Materials and Their Processing Methods

Mature specimens of acerola (*Malpighia emarginata*), guava (*Psidium guajava*), and pitanga (*Eugenia uniflora* L.) were collected between January and March 2020 in the cities of Petrolina (latitude 9°23′39″ S, longitude 40°30′35″ W, altitude 380 m) and Bonito (latitude 8°28′13″ S, longitude 35°43′35″ W, altitude 423 m), both located in the state of Pernambuco, Brazil.

The additives used were albumin (Naturaovos^®^, Salvador do Sul, Brazil), gum Arabic (Synth^®^, Diadema, Brazil), guar gum (GastronomyLab^®^, Goiania, Brazil), and gelatin (Royal^®^, São Paulo, Brazil).

The fruits were selected at the mature maturation stage, with the acerola and pitanga having completely red skins and the guava with a completely yellow skin, in addition to the absence of injuries. They were then washed in running water to eliminate foreign materials, sanitized by immersion in chlorinated water (50 ppm) for 15 min, followed by rinsing in potable water to remove the excess sanitizer.

Then, they were pulped in a horizontal batch mechanical pulping machine (Laboremus^®^, model DF-200, Campina Grande, Brazil), equipped with a sifted screen with holes of 2.5 mm in diameter. The acerola, guava, and pitanga pulps obtained were individually packaged in low-density polyethylene packages, with a capacity of 200 g, and stored in a freezer at a controlled temperature (−18 °C) until the moment of blending.

To obtain the blend, the pulps were mixed at a ratio of 1:1:1 (g:g:g), in a domestic blender (Arno^®^, Power Mix model), where they were homogenized at maximum speed for 2 min.

### 2.2. Foam Mat Drying

Three formulations were elaborated to obtain the foams. The additives albumin, gum Arabic, guar gum, and gelatin were incorporated into the red fruit blend, in the following concentrations: 6% albumin and 1% gum Arabic (formulation E1), 6% albumin and 1% guar gum (formulation E2), and 6% albumin and 1% gelatin (formulation E3). For air incorporation and foam formation, the formulations were subjected to beating in a domestic mixer (Arno^®^, Deluxe model, São Paulo, Brazil) at maximum speed (5×), with beating times of 5 min for E2 and 30 min for the other formulations.

The produced foams were distributed over stainless-steel trays, in mats with a thickness of 0.5 cm, measured with a digital caliper (Onebycitess, DD2018011001, 0–6′′/150 mm, São Paulo, Brazil) and dried in an oven with forced air circulation (FANEM, model 320E, São Paulo, Brazil) at a speed of 1.0 m/s, at temperatures of 50, 60, 70, and 80 °C. After drying, the dry material was removed with a plastic spatula, and subsequently crushed in a mini-processor (Mallory, Oggi^+^ model, Maranguape, Brazil), obtaining powders corresponding to each formulation and drying temperature.

The temperature range, the dosage of additives, and the mixing times used were previously determined through tests.

### 2.3. Characterization of Powders from Red Fruit Blend Formulations

The powdered products of the obtained blend were characterized in quadruplicate, based on the following analyses.

#### 2.3.1. Physicochemical Characterization

The physicochemical analyses of the powders were performed using the methodologies proposed by the AOAC [10], evaluating the following parameters: total titratable acidity (TTA) by titration with NaOH 0.1 mol/L to pH 8.1; total soluble solids—TSS (°Brix), in a portable refractometer (Instrutherm^®^, model RT-30 ATC, São Paulo, Brazil); water content, by drying in an oven at 105 °C (QUIMIS^®^, model Q319V) until constant mass, and mineral residue, obtained by calcination in a muffle at 550 °C. The water activity (a_w_) at 25 °C was determined by direct reading in a dew-point hygrometer (Aqualab, model 3TE, Decagon Devices^®^, Washington, DC, USA), and the pH in a digital pH meter (Tecnal^®^, model TEC-2, São Paulo, Brazil).

The total soluble sugars’ contents were determined by Yemm and Willis’ methodology [11], and for the reducing sugars’ contents, Miller’s methodology was used [12]. Both analyses were performed with a spectrophotometer (Coleman^®^, model 35 D, Santo André, Brazil).

#### 2.3.2. Bioactive Compounds

The ascorbic acid content was determined based on the protocol by Oliveira, Godoy, and Prado [13]. Total phenolic compounds (TPC) were quantified by the method described by Waterhouse [14], total carotenoids according to Lichthenthaler [15], the lycopene content according to Nagata and Yamashita [16], and flavonoids and anthocyanins according to the methodology described by Francis [17]. All absorbance readings were performed with a spectrophotometer (Coleman^®^, model 35 D, Santo André, Brazil).

#### 2.3.3. Physical Characterization

The Hausner factor was determined according to the methodology proposed by Hausner [18] and the Carr index according to Bhusari, Muzaffar, and Kumar [19]. Solubility was evaluated using the methodology of Eastman and Moore [20], modified by Cano-Chauca et al. [21].

Porosity was calculated by the relationship between apparent density and absolute density. For the evaluation of the angle of repose, the methodology described by Aulton [22] was used.

#### 2.3.4. Colorimetric Characterization

Colorimetric parameters were evaluated using a portable spectrophotometer (MiniScan, Hunter Lab XE Plus, model 4500 L, Hunter Associates Laboratory, Reston, VA, USA). Color coordinate readings were performed using the CIELAB system: L* (brightness), a* (transition from green to red), and b* (transition from blue to yellow), and hue angle (h*) and color saturation or chroma (C*) were calculated according to Equations (1) and (2), respectively:(1)$${h^{*}=\tan}^{-1}(b^{*}{/a}^{*})$$
(2)$$C^{*}=\sqrt{{(a^{*}}^{2}{+b^{*}}^{2})}$$

### 2.4. Statistical Analysis

All data were obtained and statistically evaluated using the Assistat^®^ software, version 7.7, using a completely randomized design, in a 4 × 3 factorial scheme with 4 drying temperatures (50, 60, 70, and 80 °C) and 3 formulations (E1, E2 and E3), with comparison among means using the Tukey test at 5% significance [23].

## 3. Results and Discussion

### 3.1. Physicochemical Characterization of Powder Blend Formulations

The mean values with the respective standard deviations of the physicochemical parameters of the powders of the red fruit blend formulations obtained by drying in a foam mat are presented in Table 1.

It was verified that with the increase of the drying temperature, there was a tendency of a reduction in the water contents (WC) and of the water activity (a_w_) in all the evaluated samples, with the powders presenting values of WC below 8.54 %, reaching an average of 5.97 at 80 °C and a_w_ below 0.298, results that provide safety for storage (a_w_ < 0.6 and WC < 10%). Watharkar et al. [24] reported a reduction in the water content with the increasing temperature (50, 60, and 70 °C) in the foam mat drying of ripe banana pulp (*Musa balbisiana*) added with skimmed milk powder (foaming agent), showing levels of water ranging from 7.25% to 4.87%, with the authors highlighting that the structure of the foam was more porous at higher temperatures, causing greater water loss.

Among the formulations, no consistent statistical differences were found neither in the water content nor in the water activity of the samples, considering the set of four temperatures. Variations in the water content depending on the formulation were observed by Sansomchai, Sroynak, and Tikapunya [25] when drying mango (*Mangifera indica* L.) cv. Mahachanok with different concentrations of foaming agents: glycerol monostearate (1–2%), sugar (5–10%), and soy isolate (1–3%), with values ranging from 2.58% to 4.56%.

The pH values did not significantly differ from each other (*p* > 0.05), demonstrating the absence of an effect of the formulations and temperatures on this parameter, with the powders of the formulations presenting an acidic characteristic (pH < 4.5). Powders obtained from other fruits dehydrated by foam mat drying also showed acidic characteristics, such as powders from red rose apple (*Syzygium malaccense*) dried at temperatures of 50 to 70 °C, with a pH in the range from 3.5 to 3.47 [26], and *butiá* pulp powder (*Butia* spp.) obtained at a temperature of 80 °C, with a pH of 3.25 [27].

Acidity showed greater sensitivity to changes in additives and temperatures, with the formulation containing guar gum showing greater acidity with temperatures, and the formulation with gelatin showing the lowest. Acidity contents ranging from 2.35% to 3.0% were quantified in tomato pulp powders (*Solanum lycopersicum* M.) with the addition of albumin and carboxymethylcellulose obtained from foam mat drying at temperatures of 60, 65, and 70 °C [28], and in pineapple mint powder foamed with Emustab^®^ and Super Liga Neutra^®^, obtained by foam mat drying at temperatures of 60, 70, and 80 °C [29], acidity was reported at 4.11%, 3.80%, and 2.88% (dm), respectively.

High ash contents (above 3.7 g/100 g) were found without a significant influence of the drying temperature or the type of formulation (*p* > 0.05). The similarities found in the ash content can be explained by the total proportion of additives, set at 7%. Lower values were reported in [30] in foamed red banana pulp powders with different foaming and stabilizing agents (maltodextrin, carboxymethylcellulose, acacia gum, carrageenan gum, and gelatin), obtained by foam mat drying at 60 °C, with an ash content ranging from 1.21 to 1.65 g/100 g (dm), and in avocado pulp powders (*P. americana* Mill.) obtained by foam mat drying at temperatures from 50 to 70 °C, using powdered goat milk and Emusbab^®^, with an ash content ranging from 1.25 to 2.07 g/100 g (dm) [31].

Both the increase in the drying temperature and the type of additive (gum Arabic, guar gum, and gelatin) corresponded to a proportion of 1%, significantly influencing the total and reducing sugar levels of the powders in the blend formulations. There was a trend towards a reduction in sugars with the increasing drying temperature, with the total sugars of the powdered samples of formulation E3 (containing gelatin) showing the highest values, and those of E1 (containing gum Arabic) having the lowest values; while for reducing sugars, the highest values were from E1, and the lowest were from sample E2 (containing guar gum).

From the difference between total and reducing sugars, it can be deduced that most of the total sugars were made up of non-reducing sugars. The same was observed in mixed pulp powders of jambolan and acerola, obtained by foam mat drying at temperatures from 50 to 80 °C by Matos et al. [32], in which non-reducing sugars ranged from 38.30 to 67.88 g/100 g (dm), reducing sugars ranged from 0.27 to 0.35 g/100 g (dm), and total sugars ranged between 40.67 and 69.43 g/100 g (dm).

### 3.2. Bioactive Compounds of Powder Blend Formulations

The means and standard deviations referring to the levels of bioactive compounds in the tropical red fruit blend powders, obtained by foam mat drying at temperatures from 50 to 80 °C, are presented in Table 2. 

All samples showed high levels of ascorbic acid, especially those added with gelatin, followed by those added with gum Arabic, both with a great advantage over guar gum, producing samples with about 50% of the content compared to the others. The smallest degradations in all formulations were observed at 60 °C, which can be explained by being an intermediate temperature, which combines discrete heating with a relatively short drying time. Hossain et al. [28], analyzing tomato pulp powders (*Solanum lycopersicum* M.) obtained by drying the pulp in a foam mat with albumin (3%, 5%, and 7%) and carboxymethylcellulose (0.5% and 1.0%), at temperatures of 60, 65, and 70 °C, verified the degradation of ascorbic acid with drying, with values that varied from 2.35 to 3.00 mg/100 mL, and the highest levels were also obtained at 60 °C. Ascorbic acid (AA) contents much lower than those of the evaluated powders were reported by Gonzaga et al. [29] in pineapple mint powder foamed with Emustab^®^ (2%) and Super Liga Neutra^®^ (2%), obtained at temperatures of 60, 70, and 80 °C, with values of 16.59, 33.72, and 30.29 mg/100 g (dm), respectively. Losses of AA on drying are usually high because it is one of the vitamins that is most sensitive to heating and exposure to oxygen. According to Vitor et al. [33], high temperatures and acidic pH levels can inactivate the oxidase enzyme.

The use of gelatin produced samples with the highest levels of total phenolic compounds (TPC), statistically differing from the others, while the sample added with guar gum reported the worst performance observed in AA levels. From the set of values, it can be observed that the temperature of 60 °C maintained a superior performance compared to the others, resulting in less degradation of the TPC, except for the E2 formulation, in which there were no significant differences between the temperatures. At a temperature of 60 °C, formulation E3 outperformed formulations E1 and E2 by about 15.57% and 357.49%, respectively. Even sample E2, however, can be classified as rich (contents above 500 mg/100 g) in TPC, according to Souza et al. [34].

Samples E1 and E3 surpassed the TPC contents of blueberries (*Vaccinium myrtillus*), sour cherries (*Prunus cerasus*), strawberries (*Fragaria × ananassa*), and cranberries (*Vaccinium macrocarpon*) dried at 70 °C, with TPC values of 1920 mg/100 g, 1330 mg/100 g, 3730 mg/100 g, and 1690 mg/100 g, respectively [35]; of guava pulp and guava residue, both dried at 55 °C, with a TPC content of 2500 mg/100 g and 1550 mg/100 g, respectively [36]; of dehydrated grapefruit powder in a spray dryer at 148 °C, with the addition of maltodextrin (1.25%), gum Arabic (9.4%), and isolated whey protein (1.44%), with TPC values of 1274 and 1294 mg/100 g [37], and of fig powders produced from drying fig pulp foam with albumin, carboxymethylcellulose, and maltodextrin, at temperatures of 60 to 80 °C, with TPC values between 745 and 1067 mg/100 g [6].

The powder containing gum Arabic showed, in the set of values, the highest retention of flavonoids compared to the others. The highest contents were reported at a temperature of 60 °C, with progressive reductions at higher temperatures. In powdered physalis pulp studied by Puente et al. [38], the flavonoid contents ranged from 20.83 to 24.41 mg/100 g, and as in the present study, the highest content was found in the dry powder at 60 °C.

The highest anthocyanin contents were observed in the gelatin sample, closely followed by the formulation with guar, which presented the highest value of drying at 50 °C and statistical equivalence at 70 °C, with the three formulations showing the highest contents at 60 °C. Thuy et al. [39], evaluating blackberry powders obtained by drying them in a foam mat with the addition of albumin, carboxymethylcellulose, and maltodextrin, at temperatures of 60 to 75 °C, reported anthocyanin contents ranging from 4.08 to 5.66 mg/100 g (dm), values close to those of the powders of the blend formulations studied here.

The highest degradations occurred at the lowest and highest drying temperatures, both for flavonoids and anthocyanins, which can be attributed to the longer exposure time to oxygen and heat at 50 °C and the high temperatures at 80 °C, inducing greater degradations.

The use of gum Arabic resulted in samples with the highest levels of carotenoids at all drying temperatures, with an average value between temperatures of 36.10%, surpassing the second-best average (22.92%) of the sample with gelatin, in more than 57%. The temperature of 60 °C produced the highest carotenoid contents in the three formulations. Carotenoids are natural pigments beneficial to human health. However, they are easily oxidized and are affected by the type of drying [40]. Values close to those determined in the powders of the formulations prepared with gelatin (E3) were verified in red banana pulp powders produced by foam mat drying with the addition of maltodextrin, carboxymethylcellulose, acacia gum, carrageenan gum, and gelatin, at a temperature of 60 °C, with carotenoid contents ranging from 19.99 to 21.09 mg/100 (dm) [30].

In terms of lycopene, the formulation with gelatin stood out as the stabilizing agent that ensured greater retention at the four temperatures, statistically differing from the others, except for the formulation with guar gum at a temperature of 50 °C. As with the other bioactive compounds, the highest lycopene contents of the three formulations were found in powders dried at 60 °C, proving, for such compounds, the best results in this condition.

The lower degradation of bioactive compounds in the powders dried at 60 °C compared to those dried at 50 °C can be attributed to the higher temperature being offset by the shorter drying time. This shorter duration minimizes the overall exposure to heat, thus reducing the likelihood of degradation. Additionally, the reduced drying period limits the reaction time available for the degradation of bioactive compounds, thereby aiding in preserving their integrity.

### 3.3. Physical Properties of Powder Blend Formulations

The influence of additives and temperatures on the physical properties of the powders in the formulations with the blend of tropical red fruits was evaluated using the parameters presented in Table 3. According to Santhalakshmy et al. [41], powders that present a Hausner factor (HF) < 1.2 are classified as low cohesiveness, HF between 1.2 and 1.4 are of intermediate cohesiveness, and HF > 1.4 are considered of high cohesiveness. Thus, all powders obtained from the formulations showed intermediate cohesiveness. The formulation powder with guar gum showed the highest values, indicating greater cohesiveness, while the samples containing gum Arabic had the lowest values, corresponding to lower cohesiveness. Products with close HF were found in powdered ripe banana pulp (*Musa balbisiana*) added with skimmed milk powder as a foaming agent, dried in a foam mat at temperatures of 50, 60, and 70 °C, with HF values that varied from 1.11 to 1.23, decreasing with the increasing temperature [24].

In the Carr index (CI), values of CI < 15% classify the powders as excellent fluidity, between 15% and 20% as good fluidity, between 20% and 35% as average fluidity, between 35% and 45% as poor fluidity, and CI > 45% as very poor fluidity [41]. Therefore, the powders of the formulations were classified as having poor fluidity, with formulation E1 having the lowest values, and thus, better fluidity in relation to the others. Similar fluidity was observed in [42] in blended powders of red beet (*Beta vulgaris* L. ssp. *vulgaris*), quince (*Cydonia oblonga* Miller), and cinnamon (*Cinnamomum verum*) extracts, dried in a foam mat at 60 °C with the addition of albumin (3%) and maltodextrin (0, 10%, 20%, and 30%), with CI values between 20.89% and 24.15%, except for the powder with 20% maltodextrin, which presented good fluidity (19.37%). In addition, the HF values cited by the authors also showed similarities with those found in red fruit powders, ranging from 1.24 to 1.32, demonstrating intermediate cohesiveness.

The effect of temperature on HF and CI was evident between the lowest and highest temperatures, corresponding to worsening cohesiveness and fluidity with the increasing drying temperature from 50 to 80 °C.

All powders in the red fruit blend formulations were classified as free-flowing by the angle of repose criterion, which were between 30 and 38° [43]. The free-flowing properties of powders obtained by foam mat drying depend on the air incorporated during foaming and the foaming agents and stabilizers used [44]. The increase in the drying temperature did not show a consistent influence on the angles of repose, which showed lower values in the formulation with gelatin. A similar value was reported by Huang et al. [45] studying beet powder (*Beta vulgaris*), with values of 33.59°, and higher by Ayetigbo et al. [46] in cassava powder, with values between 35.14 and 44.07°.

All powders showed solubility greater than 50%, with no obvious correlation with the increasing drying temperature. The highest solubility was observed in samples with gum Arabic, followed by those added with guar gum. High solubility is a desired property in powdered products because of the ease of reconstitution it affords them. Cól et al. [47] studied the drying of *bacaba* pulp in a foam mat at temperatures from 50 to 70 °C, reporting lower solubilities, ranging from 31.4% to 45.7%.

In the samples with guar gum, the highest porosities were determined at all temperatures, with the powders containing gum Arabic and gelatin presenting approximate results. There was a slight increase in porosity with the increases in drying temperature between 50 and 80 °C. Close values were determined by Begum et al. [48] in pineapple powders with carboxymethylcellulose, starch, and skimmed milk powder, with porosity ranging from 48.17% to 57.04%, and by Ayetigbo et al. [46] in cassava powder obtained at 80 °C, with values between 59.03% and 67.11%. High porosity implies a greater contact surface with the air, with empty spaces allowing the greater presence of oxygen, and consequently, the oxidation of sensitive compounds. On the other hand, it improves the water absorption capacity [49].

### 3.4. Colorimetric Characterization of Powder Blend Formulations

The means and standard deviations referring to the colorimetric parameters of the tropical red fruit blend powders, obtained by foam mat drying with different additives and temperatures, are shown in Table 4.

The powder containing gelatin showed higher brightness than the others, except at a temperature of 80 °C, in which the presence of guar gum resulted in higher brightness. The drying temperature variation did not show a consistent influence on L*.

The presence of gelatin also resulted in the highest redness (+a*) between the formulations, although the differences between samples with different additives were, in general, not very expressive.

The yellowness (+b*) of the sample with gelatin surpassed that of samples E1 and E2, except at 80 °C, where E2 was higher. In addition, it appears that, despite the sensory impression indicating a red color in the powders, the values indicated yellow as the predominant hue.

The highest saturation values (chroma—C*) were also present in the sample with gelatin, highlighting those determined at temperatures of 50 and 70 °C, while formulation E2 presented the highest C* in the powder obtained at 80 °C. The C* values close to zero correspond to neutral colors (gray tones), and those close to 60 indicate more intense colors [50].

Belal et al. [51], studying the drying of tomato pulp in a foam mat with albumin (3%, 5%, and 7%) and carboxymethylcellulose (1% and 0.5%), observed that with the increasing temperature (60, 65, and 70 °C), the redness (+a*) increased from 43.82 to 61.67, brightness (L*) increased from 23.13 to 28.85, and yellowness (+b*) increased from 21.29 to 26.89. Blend powders of red beet, quince, and cinnamon extract obtained by foam mat drying, added with albumin (3%) and maltodextrin (0, 10%, 20%, and 30%), showed, with the increase in the proportion of maltodextrin, elevations of L* (from 25.69 to 40.16), +a* (from 42.3 to 53.41), and +b* (from 8.87 to 10.9) [42].

The hue angles (h*) of the powders of the three formulations statistically differed from each other (*p* < 0.05) at all temperatures, without their increases showing a consistent influence. All powders had h* values between 50.75 and 65.38°, that is, between 45° and 90°, indicating a hue with a predominance of yellow with the presence of red, characteristic of orange. The powders from the E2 formulation showed the highest values of h* at temperatures of 60, 70, and 80 °C, and the powder from E3 showed the highest value at 50 °C, representing the most yellowish samples. The jambolan juice powder dried in a foam mat at 80 °C with Super Liga Neutra^®^ and maltodextrin showed a more neutral color (C* = 13.02), with a shade angle of 321.7°, i.e., with a predominance of red, but combined with a blue tint [52].

Color parameters can be influenced by combinations of temperatures and drying times, additives, pigment degradation, non-enzymatic reactions (Maillard reaction), and darkening processes, among other factors [44,51].

## 4. Conclusions

All combinations of additives used in foam mat drying of tropical red fruit blends promoted the obtainment of powders with low water and water activity contents and high levels of bioactive compounds. In addition, they did not modify the acidic characteristic of the fruits used. The obtained powders presented colors with a predominance of yellow, intermediate cohesiveness, poor fluidity, and solubility above 50%.

In general, the best temperature for obtaining powders using a combination of albumin with gum Arabic, guar gum, or gelatin was 60 °C, thus ensuring greater retention of bioactive compounds. Formulation E1, containing albumin and gum Arabic, produced powders with the highest contents of flavonoids and carotenoids, while in powders of formulation E3, combining albumin and gelatin, the highest contents of total phenolic compounds and lycopene were verified at all temperatures, in addition to the highest values of ascorbic acid and anthocyanins from 60 °C, presenting itself as the best formulation among those evaluated.

## Figures and Tables

**Table 1 foods-12-02508-t001:** Mean values and standard deviations of the physicochemical parameters of the powders of the acerola, guava, and pitanga blend formulations.

Parameters	Formulations	Drying Temperature (°C)			
		50	60	70	80
Water content ^1^	E1	8.54 ± 0.39 aA	7.48 ± 0.27 aB	6.66 ± 0.39 aB	5.02 ± 1.03 bC
	E2	7.72 ± 0.46 aA	7.25 ± 0.40 aA	7.16 ± 0.41 aA	5.75 ± 0.20 bB
	E3	8.24 ± 0.55 aA	7.70 ± 0.44 aAB	7.26 ± 0.48 aB	7.15 ± 0.35 aB
a_w_	E1	0.298 ± 0.004 aA	0.286 ± 0.002 aA	0.253 ± 0.002 abB	0.244 ± 0.001 aB
	E2	0.288 ± 0.012 aA	0.251 ± 0.010 cB	0.256 ± 0.008 aB	0.231 ± 0.010 bC
	E3	0.275 ± 0.006 aA	0.271 ± 0.005 bA	0.241 ± 0.009 bB	0.253 ± 0.006 aB
pH	E1	3.84 ± 0.02 aA	3.85 ± 0.02 aA	3.84 ± 0.02 aA	3.83 ± 0.02 aA
	E2	3.60 ± 0.03 aA	3.57 ± 0.02 aA	3.61 ± 0.08 aA	3.58 ± 0.05 aA
	E3	4.29 ± 0.04 aA	4.29 ± 0.03 aA	4.27 ± 0.02 aA	4.25 ± 0.02 aA
Total titratable acidity ^2^	E1	7.93 ± 0.14 aA	6.80 ± 0.05 bB	6.52 ± 0.11 bC	6.44 ± 0.10 bC
	E2	7.75 ± 0.17 aA	7.67 ± 0.10 aA	7.70 ± 0.26 aA	7.62 ± 0.07 aA
	E3	5.01 ± 0.7 bA	4.55 ± 0.08 cB	4.68 ± 0.19 cB	4.99 ± 0.14 cA
Ashes ^1^	E1	4.07 ± 0.11 aA	3.87 ± 0.04 aA	3.84 ± 0.09 aA	3.78 ± 0.05 aA
	E2	3.71 ± 0.18 aA	3.93 ± 0.22 aA	3.87 ± 0.16 aA	3.77 ± 0.26 aA
	E3	3.92 ± 0.15 aA	3.74 ± 0.22 aA	3.90 ± 0.29 aA	3.97 ± 0.16 aA
Total sugars ^3^	E1	15.02 ± 0.50 cB	16.09 ± 0.80 cA	14.11 ± 0.50 cC	12.24 ± 0.51 cD
	E2	23.29 ± 0.31 bA	17.76 ± 0.33 bB	15.14 ± 0.57 bC	14.06 ± 0.64 bD
	E3	24.12 ± 0.33 aA	19.99 ± 0.74 aB	17.23 ± 0.41 aC	16.72 ± 0.55 aC
Reducing sugars ^3^	E1	5.40 ± 0.14 aB	6.35 ± 0.32 aA	4.78 ± 0.34 aC	4.66 ± 0.15 aC
	E2	3.91 ± 0.11 bA	3.47 ± 0.08 bA	2.76 ± 0.09 cB	2.65 ± 0.19 cB
	E3	1.95 ± 0.09 cB	3.71 ± 0.07 bA	3.33 ± 0.09 bA	3.13 ±0.14 bA

E1—blend + albumin (6%) + gum Arabic (1%); E2—blend + albumin (6%) + guar gum (1%); E3—blend + albumin (6%) + gelatin (1%). ^1^ (g/100 g wb), ^2^ (g citric acid /100 g db), ^3^ (g/100 g db). Means followed by the same lowercase letter in columns and uppercase letter in rows did not statistically differ via Tukey’s test, at 5% probability (*p* < 0.05).

**Table 2 foods-12-02508-t002:** Mean values and standard deviations of the contents of bioactive compounds of the powders of the acerola, guava, and pitanga formulations obtained by foam mat drying.

Parameters	Formulations	Drying Temperature (°C)			
		50	60	70	80
Ascorbic acid ^1^	E1	16,343.44 ± 669.69 aC	19,358.11 ± 690.02 bA	17,950.37 ± 302.36 bB	14,595.00 ± 591.85 bD
	E2	7681.51 ± 214.11 cC	9957.15 ± 391.21 cA	8725.53 ± 222.26 cB	7645.31 ± 266.72 cC
	E3	15,598.54 ± 215.21 bD	22,198.07 ± 130.06 aA	21,217.85 ± 10,577 aB	20,390.83 ± 449.62 aC
Total phenolic compounds ^2^	E1	4090.81 ± 6.76 bB	4568.44 ± 14.98 bA	3876.26 ± 6.15 bB	2499.70 ± 485.05 bC
	E2	1280.05 ± 26.03 cA	1154.10 ± 46.21 cA	1274.04 ± 11.23 cA	1055.96 ± 28.93 cA
	E3	5053.29 ± 110.84 aAB	5279.86 ± 91.51 aA	4986.42 ± 56.61 aB	4676.28 ± 93.16 aC
Flavonoids ^1^	E1	11.45 ± 0.57 aC	13.05 ± 0.65 aA	12.45 ± 0.63 aAB	11.63 ± 0.64 aBC
	E2	7.82 ± 0.10 cC	12.43 ± 0.21 abA	10.28 ± 0.33 bB	8.50 ± 0.31 cC
	E3	9.13 ± 0.55 bC	12.14 ± 0.41 bA	11.68 ± 0.77 aA	10.27 ± 0.30 bB
Anthocyanins ^1^	E1	6.26 ± 0.22 aB	7.85 ± 0.26 bA	4.53 ± 0.14 bC	4.67 ± 0.26 cC
	E2	6.21 ± 0.30 aC	8.20 ± 0.25 bA	7.35 ± 0.12 aB	6.34 ± 0.15 bC
	E3	5.69 ± 0.23 bD	9.00 ± 0.54 aA	7.56 ± 0.28 aB	7.04 ± 0.15 aC
Carotenoids ^1^	E1	28.09 ± 1.14 aD	43.93 ± 0.67 aA	34.26 ± 1.09 aC	38.13 ± 1.25 aB
	E2	19.28 ± 0.90 bB	28.81 ± 0.94 bA	20.17 ± 0.75 cB	17.32 ± 0.41 cC
	E3	20.95 ± 0.86 bC	24.81 ± 1.66 cA	23.62 ± 1.12 bAB	22.32 ± 0.44 bBC
Lycopene ^1^	E1	0.17 ± 0.01 bC	0.24 ± 0.1 cA	0.22 ± 0.01 cB	0.18 ± 0.01 cC
	E2	0.30 ± 0.01 aA	0.32 ± 0.00 bA	0.26 ± 0.01 bB	0.24 ± 0.01 bC
	E3	0.31 ± 0.004 aC	0.35 ± 0.009 aA	0.33 ± 0.003 aB	0.33 ± 0.005 aB

Formulations: E1—blend + albumin (6%) + gum Arabic (1%); E2—blend + albumin (6%) + guar gum (1%); E3—blend + albumin (6%) + gelatin (1%). ^1^ (mg/100 g db), ^2^ (mg GAE*/100 g db). ^2^ GAE—gallic acid equivalent. Means followed by the same lowercase letter in columns and uppercase letter in rows did not statistically differ via Tukey’s test, at 5% probability (*p* < 0.05).

**Table 3 foods-12-02508-t003:** Mean values and standard deviations of the physical properties of the powders of the acerola, guava, and pitanga formulations obtained by foam mat drying.

Parameters	Formulations	Drying Temperature (°C)			
		50	60	70	80
Hausner factor	E1	1.28 ± 0.00 bB	1.25 ± 0.00 cC	1.32 ± 0.00 bA	1.32 ± 0.00 cA
	E2	1.35 ± 0.00 aB	1.40 ± 0.00 aA	1.35 ± 0.02 aB	1.40 ± 0.00 aA
	E3	1.28 ± 0.00 bB	1.28 ± 0.00 bB	1.34 ± 0.00 aA	1.35 ± 0.00 bA
Carr index ^1^	E1	22.00 ± 0.00 bB	20.00 ± 0.00 cC	24.00 ± 0.00 cA	24.00 ± 0.00 cA
	E2	26.00 ± 0.00 aC	30.00 ± 0.00 aA	26.00 ± 0.00 aC	30.00 ± 0.00 aB
	E3	22.00 ± 0.00 bC	22.00 ± 0.00 bC	25.50 ± 0.90 bB	26.00 ± 0.00 bA
Angle of repose ^2^	E1	36.44 ± 0.00 aA	36.91 ± 0.74 aA	31.61 ± 0.00 cC	35.58 ± 0.59 aB
	E2	37.15 ± 0.34 aA	35.35 ± 0.22 bB	37.90 ± 0.51 aA	35.37 ± 0.48 aB
	E3	31.59 ± 1.47 bC	32.44 ± 0.58 cB	32.89 ± 0.54 bB	33.87 ± 0.65 bA
Solubility ^1^	E1	60.51 ± 1.37 bD	93.47 ± 1.41 aA	92.05 ± 1.31 aB	76.82 ± 1.14 aC
	E2	51.44 ± 0.73 cD	71.80 ± 0.19 bC	84.53 ± 1.11 bA	75.52 ± 0.98 bB
	E3	67.35 ± 0.68 aA	65.96 ± 1.00 cC	66.42 ± 1.23 cB	64.32 ± 0.71 cD
Porosity ^1^	E1	60.45 ± 0.42 bC	57.42 ± 0.08 cD	63.47 ± 0.12 bA	62.78 ± 0.17 bB
	E2	66.93 ± 0.14 aD	68.99 ± 0.19 aC	70.05 ± 0.41 aB	72.68 ± 0.24 aA
	E3	58.98 ± 1.17 cD	62.45 ± 1.25 bB	63.03 ± 0.21 bA	61.82 ± 0.39 bC

Formulations: E1—blend + albumin (6%) + gum Arabic (1%); E2—blend + albumin (6%) + guar gum (1%); E3—blend + albumin (6%) + gelatin (1%). ^1^ (%), ^2^ (°). Means followed by the same lowercase letter in columns and uppercase letter in rows did not statistically differ via Tukey’s test, at 5% probability (*p* < 0.05).

**Table 4 foods-12-02508-t004:** Mean values and standard deviations of the colorimetric parameters of the acerola, guava, and pitanga blend powders obtained by foam mat drying.

Parameters	Formulations	Drying Temperature (°C)			
		50	60	70	80
Brightness (L*)	E1	15.79 ± 0.49 cB	18.75 ± 0.09 cA	18.24 ± 0.05 cA	14.89 ± 0.21 cC
	E2	19.80 ± 0.04 bC	17.41 ± 0.21 bD	25.08 ± 0.08 bB	30.24 ± 0.33 aA
	E3	25.50 ± 0.04 aB	19.84 ± 0.45 aD	27.16 ± 0.46 aA	21.61 ± 0.15 bB
Redness (+a*)	E1	12.67 ± 0.08 bA	9.46 ± 0.06 cB	12.93 ± 0.10 aA	12.36 ± 0.08 aA
	E2	12.57 ± 0.04 bA	11.10 ± 0.11 bB	11.54 ± 0.05 bB	10.19 ± 0.05 bC
	E3	13.30 ± 0.04 aA	12.82 ± 0.06 aB	13.10 ± 0.10 aA	12.64 ± 0.03 aB
Yellowness (+b*)	E1	15.51 ± 0.23 cB	12.05 ± 0.12 cC	17.05 ± 0.32 cA	15.68 ± 0.13 cB
	E2	17.43 ± 0.30 bC	15.36 ± 0.18 bD	21.25 ± 0.12 bB	22.24 ± 0.18 aA
	E3	22.89 ± 0.13 aA	17.28 ± 0.46 aC	22.55 ± 0.30 aA	19.34 ± 0.16 bB
Chroma (C*)	E1	20.02 ± 0.19 cB	15.33 ± 0.07 cC	21.40 ± 0.23 cA	19.96 ± 0.09 cB
	E2	21.48 ± 0.23 bB	18.95 ± 0.18 bC	24.18 ± 0.09 bA	24.46 ± 0.17 aA
	E3	26.47 ± 0.11 aA	21.52 ± 0.35 aB	26.07 ± 0.29 aA	23.11 ± 0.13 bC
Hue angle—h* (^o^)	E1	50.75 ± 0.44 cC	51.80 ± 0.41 cB	52.82 ± 0.65 cA	51.76 ± 0.35 cB
	E2	54.20 ± 0.53 bC	54.16 ± 0.36 aC	61.49 ± 0.21 aB	65.38 ± 0.16 aA
	E3	59.85 ± 0.19 aA	53.42 ± 0.84 bC	59.85 ± 0.28 bA	56.84 ± 0.23 bB

E1—blend + albumin (6%) + gum Arabic (1%); E2—blend + albumin (6%) + guar gum (1%); E3—blend + albumin (6%) + gelatin (1%). Means followed by the same lowercase letter in columns and uppercase letter in rows did not statistically differ via Tukey’s test, at 5% probability (*p* < 0.05).

## Data Availability

Data can be digitized from the graphs or requested from the corresponding author.

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
