# Peer review of "Tropical Red Fruit Blend Foam Mat Drying: Effect of Combination of Additives and Drying Temperatures"

_foods, 2023, doi:10.3390/foods12132508_

Round 1

Reviewer 1 Report

Tropical red fruits are preferred by consumers and have valuable (bioactive) components, as well. Drying is widely used for their preservation, but drying technology, and conditions (mainly temperature) has significant effect on the quality parameters (compounds, sensory properties). Foam mat drying can be adaptable for many liquid food/food raw materials. But, applicable additives (and their dosage), and the optimal values of drying conditions (temperature, time etc.) are dependent on the processed materials. Therefore, the topic of the manuscript can be considered as interesting and can provide useful information. The manuscript has a logic structure. Introduction section summarizes well the relevance of the study. Applied characterization methods (measurement of physicochemical and analytical parameters) are appropriate and described clearly. But, manuscript need revision to make it more complete and clear and improve thhe ‘visibility’ of the results.

Comments, suggestions:

In my opinion Abstract need revision to make it more complete. Please give the temperature raneg used for drying experiments, the optimum additives dosage, and the achievable water activity and loss of bioactive compounds, as well.

The role and relevance of additives for foam mat drying should be discussed in more details (with some concretized data/information) in the Introduction section.

Please give explanation how was the temperature range and additives dosage selected/determined for the experiments.

Lines 145-147 are unnecessary.

The energy efficiency of different drying conditions are not discussed (temperature-time, etc).

Details of ANOVA and real optimization of process parameters are missing.

Illustration of experimental results are poor (no figures).

Author Response

Dear reviewer
We greatly appreciate all contributions to improve our article. We would like to inform you that all the suggested corrections were carried out and those that could not be, we will take them into account for the next works.
Regarding the abstract, the temperature range was added, but it was not possible to add the other information because the abstract already has the maximum number of characters required by the journal.
The energy efficiency of different drying conditions is not discussed because a work with this objective has already been published, which is not the objective of the present work.
Due to the large number of analyzes carried out and the results corresponding to the different treatments, a 4 x 3 factorial scheme was used, involving four drying temperatures and three treatments. However, if the results were expressed in graphic form, they would contain an excessive amount of information, making it difficult to adequately describe them in tables.
Consequently, the authors believe that the way in which the data are presented allows a faster and more efficient reading, since the inclusion of this information in graphs would result in an overload of data, requiring a complete change in the statistical distribution.

Best regards

Reviewer 2 Report

This research showed that the effect of additives and drying temperatures on the powders obtained from the blend of acerola, guava and pitanga. The best temperature for obtaining powders using a combination of albumin with gum Arabic or guar gum or gelatin is 60ºC and so on.

In my opinion, this manuscript needs minor revision.

There are some drawbacks in the manuscript:

1. Line84: 1:1:1(m/m) should be revised.

2.The research content is not complex, but can add charts appropriately in research results.

English language required minor editing.

Author Response

Dear reviewer
We greatly appreciate all contributions to improve our article. We would like to inform you that all the suggested corrections were carried out and those that could not be, we will take them into account for the next works.
Due to the large number of analyzes carried out and the results corresponding to the different treatments, a 4 x 3 factorial scheme was used, involving four drying temperatures and three treatments. However, if the results were expressed in graphical form, they would contain an excessive amount of information, making it difficult to describe them adequately as in tables.
Consequently, the authors believe that the way in which the data are presented allows a faster and more efficient reading, since the inclusion of this information in graphs would result in an overload of data, requiring a complete change in the statistical distribution.

Yours sincerely

Reviewer 3 Report

Paper describes interesting data on effect of combination of 2 additives and drying temperatures of  Malpighia emarginata Psidium guajava Eugenia uniflora. 

Some minor remarks:

1. Why authors observed huge differences in acidity after drying, see E1 7,93 vs 6,44. Drying in so mild conditions should not affect on acid content in plant material;

2. Why authors observed huge differences in sugar content  after drying, see e.g. E2 23,29 (50 oC) vs 15,14 g/100g (70 oC)? Drying in so mild conditions should not affect on sugar  content in plant material;

3. Why ascorbic acid amount increases when you dry in 60oC (7681,51) in comparision to 50 (9957,15±391,21)? Also in other cases?

4. Why Lycopene amount increases when you dry in 60oC (0,17 ± 0,01) in comparisaion to 50 (0,24 ± 0,1)?

5. How pH was measured in dried fruits?

Author Response

Dear reviewer

We greatly appreciate all contributions to improve our article. We would like to inform you that all the suggested corrections were carried out and those that could not be, we will take them into account for the next works.

The information on how the pH was measured is already found in the work: using the AOAC methodology and measured in a digital pH meter (Tecnal®️, model TEC-2).

Questions related to acidity and sugars can be answered taking into account the high porosity of foamy materials when subjected to drying. Greater contact with oxygen can cause greater reactions in foam layer drying.

We are available to answer any other queries.

Best regards

Round 2

Reviewer 1 Report

The manuscript has an interesting topic that has practical relevances. Authors have revised the manuscript thoroughly according to reviewers' comments and suggestions and provided detailed answers for reviewers' questions. Revision made the manuscript more complete and clear. The overall scientific quality of the MS has beein improved significantly due to the revision. I agree and accept all modifications made by the authors.